# Recent Advances in Catalytic Alkyne Transformation via Copper Carbene Intermediates

**DOI:** 10.3390/molecules27103088

**Published:** 2022-05-11

**Authors:** Kuiyong Dong, Mengting Liu, Xinfang Xu

**Affiliations:** 1School of Chemistry and Materials Science, Jiangsu Normal University, Xuzhou 221116, China; 2Guangdong Provincial Key Laboratory of Chiral Molecule and Drug Discovery, School of Pharmaceutical Sciences, Sun Yat-sen University, Guangzhou 510006, China; liumengtingxx1999@163.com

**Keywords:** copper catalysis, carbene intermediate, alkyne functionalization, cross-coupling reaction, cyclopropenation, carbene/alkyne metathesis

## Abstract

As one of the abundant and inexpensive metals on the earth, copper has demonstrated broad applications in synthetic chemistry and catalysis. Among these copper-catalyzed advances, copper carbenes are versatile and reactive intermediates that can mediate a variety of transformations, which have attracted much attention in the past decades. The present review summarizes two different reaction models that take place between a copper carbene intermediate and alkyne species, including the cross-coupling reaction of copper carbene intermediate with terminal alkyne, and the addition of copper carbene intermediate onto the C–C triple bond. This article will cover the profile from 2010 to 2021 by placing emphasis on the detailed catalytic models and highlighting the synthetic applications offered by these practical and mild methods.

## 1. Introduction

Transition-metal-catalyzed transformations of carbon-carbon triple bond have been presented as one of the most effective and prominent tools for the construction of functionalized molecules and fine chemicals [1,2,3,4]. Alkynes have low C_sp_–H p*K*a (~25), and the carbon-carbon triple bond could be selectively activated by π-acidic transition metals. These unique characteristics have rendered nucleophilic and/or electrophilic properties of alkynes, which make them versatile building blocks in the synthesis of natural products and biologically active molecules [5,6]. In this area, only a few metal complexes based on late transition metals such as platinum and coinage metals could enable the activation of alkynes [7,8,9].

On the other hand, metal carbenes, which are generated in situ via transition-metal-catalyzed transformations from different carbene precursors, are powerful reaction intermediates in organic synthesis [10,11,12,13]. Catalytic functionalization of alkynes with metal carbene species has attracted substantial attention in both academic research and industry during the past decades [14,15,16,17,18,19]. In comparison to the catalytic transformations with precious metal catalysts (e.g., gold, rhodium, and platinum complexes) [20,21,22,23,24], the use of copper catalysts is much more appealing, either for the formation of a copper carbene intermediate or for the activation of alkyne species, because of its lower cost, less toxicity, and easier accessibility. The typical transformations in this area include alkynylation [25,26,27,28,29], cycloaddition [30,31,32,33,34], allene formation [35,36,37,38,39], and many others [40,41,42,43,44].

The copper carbene species, first identified by Roy et al. in 1906 [45], were generated from the decomposition of ethyl diazoacetate in the presence of copper dust above 80 °C heating conditions. However, the development in this area was sluggish during the last century (Figure 1). Until 2000, with abundant copper salts such as Cu(acac)_2_, Cu(hafaca)_2_, 

CuOTf, Cu(OTf)_2_, CuPF_6_ (see Appendix A, Table A1 for details), and others, suitable for the development of copper carbene chemistry, a rapid growth has been achieved in this area [46,47,48]. In addition, the advancement made in modern era ligand design further accelerated the progress of copper carbene chemistry [49,50,51,52,53,54,55,56,57,58], especially in catalytic asymmetric versions. A variety of stereoselective reactions have been realized with copper carbene intermediates, including X–H insertion [59,60,61,62,63,64], cyclopropanation [65,66], cycloaddition [67,68,69], ylide formation [70,71,72], and others [73,74]. The representative advances in this area have been summarized by Doyle [75,76,77], Zhou [78,79], Wang [80,81,82], Pérez [83,84], and Davies [85,86]. However, no topic review article on the copper carbene reaction with alkyne has been reported so far. This review article will summarize and discuss two distinct reaction modes. Firstly, the cross-coupling reaction of a copper carbene intermediate with terminal alkynes delivers alkynoate or allenoate copper intermediates, and each of these intermediates could be followed by protonation, nucleophilic substitution, electrophilic addition, or elimination process, yielding functionalized alkynes (Figure 1, paths a and b) and allenes (Figure 1, paths c and d), respectively. Secondly, the addition of a copper carbene intermediate onto the C–C triple bond, which involves cyclopropenation or carbene/alkyne metathesis process, forms complex molecules with structural diversity through cascade transformations (Figure 1, paths e and f). This article will cover the profile from 2010 to 2021 by placing emphasis on the detailed catalytic models and highlighting the synthetic applications offered by these practical and useful methods.

## 2. Cross-Coupling Reaction of Copper Carbene Intermediate with Terminal Alkyne

Copper-catalyzed cross-coupling reaction of a copper carbene intermediate with terminal alkynes was one of the most powerful protocols for the construction of C–C bonds [87]. However, in early works, a mixture of alkynoates and allenoates was generated in combined moderate yields under harsh reaction conditions [88]. Until 2004, Fu reported the first example of the copper-catalyzed coupling reaction of terminal alkynes with diazo esters or diazo amides to yield 3-alkynoate or 3-butynamide products selectively with minimal amount of allene byproducts under no-basic conditions [89]. Consequently, a variety of copper-catalyzed coupling reactions of terminal alkynes with various carbene precursors have been developed independently.

### 2.1. Alkynylation

#### 2.1.1. Alkynylation Terminated by Protonation

In 2012, Wang reported a copper-catalyzed cross-coupling of *N*-tosylhydrazones **1** with trialkylsilylethynes **2**, leading to the alkynylated products **4** via the formation of C(sp)–C(sp^3^) bonds (Figure 2). Mechanism study shows that migratory insertion of copper carbene species gives the alkynoate copper intermediate **3**, and sequential protonation affords the target products **4**. This coupling reaction proceeded efficiently with *N*-tosylhydrazones derived from aromatic and aliphatic aldehydes or ketones in moderate to excellent yields without detecting the formation of allene byproducts **5**. However, when a *tert*-butyl substituted alkyne **7** was employed, the corresponding allene product **8** was formed selectively [90].

Later in 2014, Zhou reported a copper-catalyzed coupling reaction using dialkoxycarbenes **9** as carbene precursor [91], which provides unsymmetrical propargylic acetals **11** in moderate to good yields (Figure 3).

In 2018, an asymmetric coupling reaction of *N*-tosylhydrazones **12** with terminal alkynes **13** was achieved by Uozumi and co-workers using chiral copper(I)/phosphoramidite complex as the chiral catalyst (Figure 4), and optically active alkynylated product **14** was generated in moderate to good yields and enantioselectivities [92].

#### 2.1.2. Alkynylation Terminated by Electrophilic Addition

Beyond the copper-catalyzed alkynylation terminated by protonation, the alkynoate copper intermediates, formed in situ from copper carbene species and terminal alkynes, could be intercepted through a nucleophilic substitution or electrophilic addition process. In 2015, Wang and co-workers contributed a three-component cross-coupling reaction of terminal alkyne with *α*-diazo ester and alkyl halide or Michael acceptor [93]. In this transformation, *α*-diazoesters **15** react firstly with the (triisopropylsilyl)acetylene **16** through a migratory insertion process to form the alkynoate copper intermediate **17**, followed by a nucleophilic substitution with alkyl halides **18** or Michael addition with electron-deficient alkenes **20** to produce the three-component products **19** and **21**, respectively (Figure 5). This transformation represents a highly efficient method for the construction of alkynylation products with an all-carbon quaternary center in moderate to high yields. Notably, the copper catalyst works as the only catalyst to install two new C–C bonds on the carbenic carbon in this reaction.

In 2018, Hu and co-workers reported a copper-catalyzed three-component [1+2+2]-cycloaddition of trifluoromethyl diazo compounds **22** with terminal alkynes **23** and nitrosoarenes **24** [94]. With this method, a series of trifluoromethyl-substituted dihydroisoxazoles **26** were obtained in high yields under mild conditions. Mechanistically, electrophilic trapping of the alkynoate copper intermediate by nitrosobenzenes was proposed as the key step in this cascade transformation, which forms a proposed intermediate **25**, followed by a copper catalyzed intramolecular annulation to deliver the target products **26** (Figure 6).

One year later, the same group reported a copper-catalyzed three-component reaction of terminal alkynes **10** with *α*-diazoamides **27** and isatin ketimines **28** in 2019 [95]. A series of alkynyl-containing 3,3-disubstituted oxindoles **30** were efficiently formed in high yields and diastereoselectivities through a Mannich type trapping of an in situ generated alkynoate copper intermediate **29** (Figure 7).

#### 2.1.3. Alkynylation Terminated by β-Elimination

Besides the intermolecular trapping reactions of the alkynoate copper intermediate, *β*-H elimination could occur with this reactive species, providing a variety of conjugated enynes. Representative advances in this area have been reported by Wang’s group [96,97,98]. In 2015, they reported a copper-catalyzed cross-coupling reaction of terminal alkynes **9** with trifluoromethyl ketone *N*-tosylhydrazones **31**, which provides an efficient synthesis of 1,1-difluoro-1,3-enyne derivatives **33** under mild reaction conditions [96]. Mechanistically, the alkynoate copper intermediate **32** was generated in situ through a migratory insertion of the copper carbene intermediate, followed by *β*-F elimination, leading to the *gem*-difluoroolefination products **33** in moderate to high yields [Figure 8, Eq. (a)]. Later, the author reported a copper-catalyzed three-component reaction of a (triisopropylsilyl)acetylene **16** with Ethyl diazoacetate **34** (EDA) and aldehydes **35** that provided an efficient method for the synthesis of *α*-alkynyl-*α*, *β*-unsaturated esters **37** [97]. In this cascade reaction, nucleophilic aldol addition of an alkynoate copper intermediate with aldehyde **35** formed product **36**, which delivered the desired products **37** as a single (*E*)-stereoisomers through an elimination process in good to excellent yields [Figure 8, Eq. (b)]. In the same year, an analogous cross-coupling reaction with *α*-diazo phosphonates **38** instead of EDA through a sequential alkynylation/aldol addition/Horner−Wadsworth−Emmons (HWE) type reaction was disclosed by the same group [98]. This method provided straightforward access to conjugated enynes **40** with good stereoselectivity and excellent functional group compatibility [Figure 8, Eq. (c)]. Moreover, one C–C bond and one C=C bond were formed successively in a one-pot manner, making those novel enynes synthesis methods practically useful.

### 2.2. Allenylation 

#### 2.2.1. Allenylation Terminated by Protonation

As a complementary to Fu’s method for the selective synthesis of alkynoates [89], in 2011, Fox’s group reported a selective coupling reaction of *α*-substituted-*α*-diazoesters **41** with terminal alkynes **10** to the syntheses of allenoates in the presence of Cu(II)(trifluoroacetylacetonate)_2_ and 3,6-di(2-pyridyl)-*s*-tetrazine **L2** in DCE [99]. As a result, allenoates **43** were obtained as the main products with slight traces of the alkynoates. Key to the development of this selective method was the recognition of an adventitious base, potassium carbonate, which improved the selectivity of isomerization to form the allenoate products (Figure 9).

Later in 2013, an efficient copper-catalyzed cross-coupling between diazoacetamides **44** and terminal alkynes **10** under ligand-free conditions was developed by Sun [100]. This method provided a practical method for the assembly of substituted 3-butynamides **45** and dienamides **46**. Interestingly, when sodium carbonate was added to the reaction mixture, the allenic compounds **46** were obtained as the main products. However, alkyne products **45** were generated as the major products in the absence of this base (Figure 10). Moreover, the alkynoate compounds **45** could be smoothly converted into the isomeric allenes **46** in the presence of sodium carbonate without assistance of the copper catalyst.

In the same year, Wang and co-workers developed a series of synthetic methods to form substituted allenes under optimized conditions in the presence of copper(I) complexes [101,102,103,104]. Diffident types of substituted diazo compounds and *N*-tosylhydrazones were employed as the carbene precursors for the coupling with various terminal alkynes, delivering the allenoic derivatives in good yields with a wide range of functional group tolerance (Figure 11). Notably, ethyne **54** was also a compatible substrate for this reaction, which leads to a new synthetic method for the synthesis of terminal allenes **55** in moderate to excellent yields [104]. However, using one equivalent amount of CuI with DMF as the solvent is critical to the success of this transformation [Figure 11, Eq. (d)].

In 2015, a copper-catalyzed coupling reaction between flow-generated unstabilized diazo compounds and terminal alkynes was reported by Ley’s group, providing a practical method for the synthesis of di- and tri-substituted allenes **59** in high yields under mild conditions [105]. The unstable diazo compounds **57** were generated in situ from hydrazones **56** through oxidation with activated MnO_2_. Then, the above solution was injected directly into the other reaction mixture, which contained terminal alkynes **58**, base, and CuI catalyst. The reaction delivered the allene products **59** in good to excellent yields. To highlight the selectivity and functional group compatibility of this protocol, norethindrone and propargylated quinine were successfully applied to the optimal reaction conditions, generating the corresponding products in 63% and 82% yields, respectively (Figure 12).

In addition to the diazo compounds, conjugated eneyne ketones **60** were introduced as carbene precursors by Wang and co-workers in 2016 in a copper-catalyzed cross-coupling reaction with terminal alkynes [106]. This reaction afforded trisubstituted allenes **61** in high yields with broad functional group tolerance under mild reaction conditions (Figure 13).

In 2015, Feng and Liu contributed an asymmetric cross-coupling of *α*-diazoesters **15** with terminal alkynes **10** using chiral Cu(I)/ guanidine complex as the catalyst [107]. Notably, no additional base was necessary for this transformation, providing optically active 2,4-disubstituted allenoates **62** under mild reaction conditions in good to high yields (up to 99 %) with good to excellent enantioselectivities (Figure 14).

In 2016, Wang and co-workers reported a highly enantioselective copper-catalyzed cross-coupling of aryldiazoalkanes **57** with terminal alkynes **10** [108]. By utilizing chiral Cu(I)/bisoxazoline ligand **L4**, this reaction delivered a series of trisubstituted allenes **63** in moderate to high yields (up to 96%) with excellent enantioselectivities (up to 98% *ee*). Unlike the previous works using CuI as the catalyst, Cu(MeCN)_4_PF_6_ complex was used as the optimal metal catalyst to enable high reactivity and stereoselectivity in this reaction (Figure 15). 

One year later, Ley’s group reported their continuous flow strategy for the asymmetric coupling reaction of unstabilized diazo compounds **65** with propargyl amines **66** in the presence of chiral Cu(I)/PyBIM complex [109]. This method generated the amino-substituted chiral allenoates **67** in moderate yields (up to 57%) with high enantioselectivities (up to 96% *ee*) in a fast reaction rate (10–20 min) with a variety of functional group compatibility (Figure 16).

#### 2.2.2. Allenylation Terminated by Electrophilic Addition

In 2018, a one-pot copper-catalyzed asymmetric three-component reaction of diazoesters **15** with terminal alkynes **10** and isatins **68** was reported by Liu’s group [110]. Axially chiral *tetra*-substituted allenoates **70** bearing a stereogenic center were obtained under a chiral Cu(I)/guanidinium salt/YBr_3_ catalytic system with high diastereo- and enantioselectivities. In this work, the aldol type addition of allenoate-copper intermediates **69** with isatins **68** has been proposed as the key step in this reaction. Moreover, convincing experimental evidence for the formation of allenoate-copper intermediate **69** was provided through the synthesis of chiral allenoate, which was generated from the C–H insertion reaction of α-diazoester with alkyne. The author found that additional acids improved the catalyst efficiency of the chiral copper complex. The intramolecular nucleophilic trapping reaction of allenoate-copper intermediate with embedded aldehyde species was also successful, generating the cyclic allenoate product **73**, albeit the yield and stereoselectivity were moderate (Figure 17).

Recently, Sun’s group has realized an enantioselective intramolecular nucleophilic aldol addition of in situ formed allenoate-copper intermediate with aldehyde using chiral Cu(II)/Box complex [111]. Distinct from the previous version with copper(I) catalysts, this protocol used copper(II) salt as an optimal catalyst in this asymmetric cross-coupling reaction. The *tetra*-substituted allenoates **75** containing both central and axial chiralities have been obtained in moderate to good yields with good to excellent stereoselectivities. (Figure 18). 

#### 2.2.3. Allenylation Terminated by Allylation

In 2016, Wang and co-workers realized the synthesis of allyl-substituted allenes through trapping of allenoate-copper intermediate with allyl bromide through a nucleophilic substitution process [106]. In this reaction, conjugated eneyne ketones **60** have been used as the carbene source. Mechanistically, the cooper-(2-furyl) carbene intermediate was generated in situ from eneyne in the presence of CuI, followed by a migratory insertion process to afford nucleophilic alkynoate copper intermediate **77** that was trapped by allyl halide **76** (Figure 19). In this method, the choice of the base was pivotal for the reaction outcomes when K_2_CO_3_ was employed as the base, affording 2-furyl substituted allenes **78** in generally good yields.

#### 2.2.4. Cascade Transformations Involving Allenylation Process

Nucleophilic addition with allenoic ester or its isomeric compound is one of the generally used synthetic strategies for the expeditious construction of highly functionalized carbocycle or heterocycle structures [112,113,114,115,116,117,118,119,120]. Thus, a variety of inter- or intra-molecular cascade reactions have been developed through different nucleophilic addition processes of the allene derivatives that were generated from cross-coupling between alkynes and copper carbenes.

In 2011, a one-pot synthesis of phenanthrenes **81** via ligand-free CuBr_2_-catalyzed coupling reaction/intramolecular cyclization of terminal alkynes **23** with *N*-tosylhydrazones **79** derived from *o*-formyl biphenyls was developed by Wang and co-workers [121]. In this cascade reaction, allene intermediates **85** were initially generated through a cross-coupling reaction of *N*-tosylhydrazones **79** with terminal alkynes **23**, followed by a 6π-cyclization and isomerization to deliver the phenanthrene products **81** with broad functional group compatibility (Figure 20). 

Later in the same year, instead of using *o*-aryl substituted *N*-tosylhydrazones, *o*-hydroxy- or *o*-amino-substituted *N*-tosylhydrazones were introduced by the same group as carbene precursors in an analogous cascade transformation, a ligand-free CuBr-catalyzed coupling reaction/intramolecular cyclization sequence, achieving the synthesis of benzofuran or indole derivatives **84** in moderate to excellent yields [122]. The initially formed allene intermediates **83** were trapped through a nucleophilic addition by the embedded *o*-hydroxy- or *o*-amino group to afford the cyclized products **84** (Figure 21).

In 2011, a similar catalytic strategy was developed by Balakishan’s group. They reported a simple procedure for the synthesis of aza- and oxa-cycles via a copper-catalyzed coupling reaction of functionalized terminal alkynes **85** with diazoesters **86** [123]. Initially, the allene intermediates were formed in the presence of CuI, followed by an intramolecular aza- or oxa-Michael cycloaddition and isomerization to generate the cyclized five- or six-membered products **87** in generally good yields (Figure 22).

In 2015, a stereo-divergent synthesis of five-membered heterocycles was developed by Sun’s group [124]. This work described a copper-catalyzed cross-coupling reaction and annulation cascade reaction of amino alkynes **88** with diazo compounds **15**. The proposed reaction mechanism involves trapping in situ formed allene intermediates, yielding 2-methylenes **89** (when PG = Bn) and 2,3-dihydropyrroles **90** (when PG = Ts) in good yields with broad functional group tolerance under mild conditions. Control experimental results show that *N*-benzyl amino alkynes were more likely to form 2-methylenespyrroles derivatives **89** through *5-exo-dig* cycloaddition, while 2,3-dihydropyrroles **90** generated from *N*-tosylamino alkynes through *5-endo-dig* cycloaddition would be more favorable (Figure 23).

In 2018, Sun and co-workers expanded the above chemistry to the synthesis of the four- to six-membered heterocycles with *N*-substituted prop-2-yn-1-amines **91** and diazoacetates **15** [125]. Generated allenoic species **92** have been proven as the key intermediates for the subsequent diverse annulations under optimized conditions toward functionalized heterocycle in moderate to good yields. Treatment of allenoates **92** with sodium phenolates led to six-membered products **93**; silver nitrate and triethylamine yielded five-membered products **94**; and what’s more, four-membered products **95** were generated under lithium *tert*-butoxide conditions (Figure 24).

In addition to the cyclization through addition with a heteroatom, carbon-based nucleophilic species could also be served as the nucleophile to addition with these allenes, forming the C–C bond instead of the C–X bond [126,127]. In 2015, Kumaraswamy’s group developed a cooper catalyzed cross-coupling reaction/intramolecular Michael addition cascade reaction [128], achieving the formation of indene and dihydronaphthalene derivatives **97** in good yields with broad functional group tolerance [Figure 25, Eq. (a)]. Later in 2017, Sun’s group reported an analogous approach toward five- or six-membered carbo-/heterocycles with diazo compounds **15** and alkyne-substituted malonates **98** [129]. In this reaction, the ligand significantly enhanced the reaction yields and inhibited the Conia-ene side reaction. As a result, the polyfunctionalized cyclohexenes, tetrahydropyridines, and dihydropyrans have been prepared in moderate to high yields under mild reaction conditions [Figure 25, Eq. (b)].

In 2015, a Cu(I)-catalyzed denitrogenative annulation reaction of pyridotriazoles **100** with terminal alkynes **10** was developed by Gevorgyan’s group [130]. Initially, *α*-pyridyl copper carbenes were generated from pyridotriazoles **100** in the presence of the copper catalyst, followed by a cross-coupling reaction with terminal alkyne to form either propargylic or allenoic intermediates **101**, which were terminated by copper-catalyzed cycloisomerization to furnish the indolizines **102** in moderate to excellent yields (Figure 26).

In 2018, Wang and co-workers reported a copper-catalyzed geminal difunctionalization reaction of terminal alkynes [131]. The key step in this cascade reaction is trapping the in situ generated allenoic species **105** with a sulfonyl anion to form the carbon-sulfur bond, providing a variety of vinyl sulfones **106** in good yields with excellent stereoselectivities under mild reaction conditions. It was noted that the excellent stereoselectivities might be due to the influence of steric hindrance, and no ligand and additive was required in this transformation (Figure 27).

Recently, Sun and co-workers have demonstrated a copper-catalyzed three-component reaction of terminal alkynes with diazo compounds and B_2_pin_2_ for the synthesis of trisubstituted alkenylboronates [132]. In this alkyne difunctionalization transformation, the copper catalyst plays dual roles in the whole process. Initially, copper catalyzed the cross-coupling to form an allenoic intermediate, followed by a copper-catalyzed stereoselective boration reaction with B_2_pin_2_. When diazo compounds **53** were used as carbene precursors, the steric interaction forced the boron group to attack the *β*-carbon from the opposite side of the *γ*-phenyl group on the allenoic species **107**, leading to the favored (*Z*)-isomers **108** as major products. Whereas, in the case with *N*-tosylhydrones **51** as carbene precursors, the addition of Cu-Bpin complex to corresponding allenoic species **109** provided allyl copper intermediate, which was more favored to form a six-membered ring transition state with the association of MeOH, finally furnishing the more thermodynamically stable (*E*)-products **110** (Figure 28).

## 3. Copper Carbene Intermediate Addition onto C–C Triple Bond

### 3.1. Cyclopropenation

Cyclopropenation is a well-known reaction of metal carbene intermediate with alkynes. This widely used reaction could be catalyzed by rhodium [133,134,135,136,137], cobalt [138], gold [139], silver [140,141] and many others [142,143,144]. Herein, selected examples related to copper catalysis will be discussed. 

In 2010, a new tridentate coordination copper complex, Cu[Ms(CH_2_SCN)_3_]BAr’_4_ (BAr’4 = tetra(3,5-bis(trifluoromethylphenyl)borate), was designed by Miguel and co-workers by using [Cu(OTf)]_2_•C_6_H_6_ and an alkylthiocyanate ligand [145]. This catalyst promoted the cyclopropenation of ethyl diazoacetate **34** (EDA) with a wide range of internal alkynes **111**, providing cyclopropenes **112** in moderate yields (Figure 29). The same cyclopropenation work was achieved by Dias, in which unique bis(pyrazolyl)borate ligand supported [(CF_3_)_2_Bp]Cu(NCMe) catalyst was used, yielding cyclopropene products in moderate to high yields [146].

In 2016, a Cu(I)/*N*-heterocyclic carbene (CuNHC) complex catalyzed cyclopropenation of internal alkynylsilanes **113** with diazoacetate **15** was reported by Coleman’s group [147]. A series of 1,2,3-trisubstituted and 1,2,3,3-tetrasubstituted cyclopropenylsilane compounds **114** were isolated in moderate to good yields (Figure 30). An interesting regioselective and chemodivergent reaction pathway occurred to furnish a *tetra*-substituted furan through an intramolecular cyclopropane and ring-opening cascade process in the case of electron-rich diazoacetate.

### 3.2. Cascade Reaction Involving Carbene/Alkyne Metathesis Process

Carbene/alkyne metathesis (CAM) refers to the processes where a metal carbene reacts with an alkyne, generating a new vinyl metal carbene intermediate, which was difficult to access with other carbene precursors [148,149,150]. This in situ generated vinyl metal carbene intermediate could be involved in typical metal carbene reactions, such as [3+2]-cycloaddition [151], cyclopropanation [152,153,154], C–H bond insertion [155,156,157,158], and others [159,160,161]. Herein, we summarized recent works on the copper-mediated cascade transformations involving carbene/alkyne metathesis.

It is a general protocol for the synthesis of furan derivatives through transition metal-catalyzed formal [3+2]-cycloaddition of *α*-diazocarbonyl compounds with alkynes [162,163,164,165]. However, the cases under copper carbenes mediated were limited. In 2014, Wang’s group developed a copper-catalyzed formal [3+2]-cycloaddition reaction of terminal alkynes with *β*-keto *α*-diazoesters **115** (X = O), offering an operationally simple and applicable method for the synthesis of trisubstituted furans **116** (X = O) with a wide substrate scope [Figure 31, Eq. (a)] [166]. This reaction has also been applied to ethyl (*E*)-2-diazo-3-(methoxyimino)butanoate **115** (X = NOMe) for the synthesis of 2,3,5-trisubstituted *N*-methoxypyrroles (X = NOMe). Later in 2016, a Cu(I)-catalyzed cycloaddition of diazoacetates **15** with electron-rich internal aryl alkynes **117** was discovered by Coleman and co-workers [167]. Tetra-substituted furans **118** were generated in moderate isolated yields with high chemoselectivities and regioselectivities [Figure 31, Eq. (b)].

In 2016, Xu’s group developed a chemo-divergent copper-catalyzed cascade reaction of alkynyl-tethered *α*-iminodiazoacetates **119**, providing polycyclic and multi-substituted pyrroles in high yields with a broad substrate scope [168]. Especially, the *tetra*-substituted 3-formylpyrroles **124**, which were difficult to access by alternate approaches. Mechanistic studies indicated that the *α*-imino carbene **120** is the key common intermediate in this divergent reaction, which was generated by metal-catalyzed carbene/alkyne metathesis of the alkynyl-tethered diazo compounds **121**. When R^1^, R^2^ was imbedded with an aromatic ring, polycyclic pyrroles **122** were formed as the major products through a [3+2]-cyclization and aromatization process. Whereas, for substrates with a methoxy group on the nitrogen (R^2^ = OMe), the carbene intermediate underwent an *N*–O insertion/alkoxy migration/alcoholysis sequence, giving the 3-formylpyrrole products **124** in generally good to excellent yields (Figure 32).

At the same time, Xu and co-workers also developed a copper-catalyzed carbene/alkyne metathesis cascade reaction with alkyne-tethered diazo compounds **125** [169]. This transformation provided a rapid access for the construction of multi-substituted 4-carboxyl quinoline derivatives **127** in high to excellent yields. In this cascade reaction, one C=N and one C=C bond were formed with the assistance of the copper catalyst under mild reaction conditions [Figure 33, Eq. (a)]. Later in 2017, Ye’s group reported an analogous protocol by using ynamides **128** as carbene precursor [170]. In this work, a copper carbene was generated in situ through a catalytic oxidation process in the presence of quinoline *N*-oxide, followed by a CAM process and terminated by carbene reaction with an embedded azide group, providing a wide range of pyrrolo [3,4-*c*]quinolin-1-ones **130** in good yields [Figure 33, Eq. (b)]. These works represented a practical method for the dual functionalization of alkynes.

In addition to the nucleophilic addition of the in situ formed copper carbene intermediates, electrophilic aromatic substitution or C(sp^2^)–H bond functionalization is another useful terminating transformation for the direct construction of polycyclic fused frameworks. In 2017, Doyle’s group reported a copper-catalyzed intramolecular cascade reaction of diazo compounds **131**. This transformation went through a CAM process followed by a carbene C(sp^2^)–H bond functionalization cascade, yielding the fused indeno-furanone derivatives **133** in excellent yields under mild reaction conditions [Figure 34, Eq. (a)] [171]. Instead of terminating the reaction through C–H functionalization, a selective Buchner insertion reaction occurred as the terminating step in Xu’s work when the *ortho*-aniline substituted propargyl diazoacetates **134** were employed, selectively affording the dihydrocyclohepta[*b*]indole derivatives **136** in moderate to high yields [Figure 34, Eq. (b)]. Notably, this reaction described a rare example of the Buchner reaction with donor/donor type metal carbene species [172]. 

In 2018, Xu and co-workers developed an intermolecular copper-catalyzed formal CAM process [173], which underwent a copper promoted [3+2]-cycloaddition/dinitrogen exclusion/nucleophilic addition process, providing a direct and effective access to 2*H*-chromene derivatives **139** in generally good to high yields. Mechanistic studies indicated that the 3*H*-pyrazole **138** is the key intermediate in this cascade transformation, and this critical intermediate was isolated and confirmed by single-crystal X-ray diffraction and spectroscopy analysis for the first time (Figure 35).

Based on a similar protocol, a copper-catalyzed formal [1+2+2]-annulation of alkyne-tethered diazo compounds **140** with pyridines **141** has been reported by Xu’s group recently [174]. In contrast to the previously reported cascade reaction that terminated the copper carbene intermediate on the carbonic center, a vinylogous addition of vinyl carbene intermediate with pyridine derivatives occurred in this reaction, followed by an intramolecular annulation to form cycloadducts **146**, which underwent a decarboxylative aromatization process to form the desired polycyclic fused indolizine derivatives **147** in good to high yields (Figure 36, path a); although, direct formal [3+2]-cycloaddition *via* pyridinium ylide pathway could not be ruled out so far (Figure 36, path b).

## 4. Conclusions

This review has summarized the recent progress of catalytic alkyne functionalization involving copper carbene intermediates. Copper carbene species derived from different carbene precursors have been introduced to react with alkynes through two distinguished reaction models: the cross-coupling reaction of copper carbene intermediates with terminal alkynes and the addition of copper carbene intermediates onto the C–C triple bond. The former reaction involves alkynoate or allenoate copper intermediates, followed by protonation, nucleophilic substitution, electrophilic addition, or elimination process, yielding functionalized alkynes and allenes, respectively. The latter version includes cyclopropenation and cascade reaction via carbene/alkyne metathesis process. Although substantial progress has been realized in this field, challenges remain, e.g., the carbene precursors are still mainly limited to the diazo compounds; the catalytic efficiency could be further improved, especially in the asymmetric catalysis; and synthetic applications of this chemistry are still under exploration. Therefore, the synthetic potential of this chemistry could be envisioned through the introduction of a variety of readily accessible carbene precursors and with the development of robust copper catalysts/ligands, including novel methodology discovery for the catalytic alkyne functionalization, and expeditious assembly of molecules with structural complexity and diversity for the leading compound development.

## Data Availability

Not applicable.

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
