# Peer review of "Recent Advances in Catalytic Alkyne Transformation via Copper Carbene Intermediates"

_molecules, 2022, doi:10.3390/molecules27103088_

Round 1
Reviewer 1 Report
The following corrections need make and issues need address.
- The all Schemes need to be homogenized throughout the manuscript.
- Make sure that any abbreviations that are used are defined somewhere in the manuscript.
- There are some grammatical errors that need to be corrected. I, therefore, recommend that the authors contact an English speaker or somebody fluent enough to help them correct the paper as it needs complete revision.
Author Response
For Reviewer 1:
Comments and Suggestions for Authors
The following corrections need make and issues need address.
- The all Schemes need to be homogenized throughout the manuscript.
Response: We thank the reviewer for the comment. All the Schemes were checked and revised throughout the manuscript.
- Make sure that any abbreviations that are used are defined somewhere in the manuscript.
Response: We thank the reviewer for pointing out. Additional table was added in the revised manuscript for the abbreviations
- There are some grammatical errors that need to be corrected. I, therefore, recommend that the authors contact an English speaker or somebody fluent enough to help them correct the paper as it needs complete revision.
Response: We thank the reviewer for the comment. The manuscript has been carefully checked for the grammatical errors.
Reviewer 2 Report
General comments:
The manuscript by Dong et al. describes two specific reactions of copper carbene species with alkynes, which includes their addition or cross couplings to carbenes. Since the topic presented in this review has not appeared in the literature before, as claimed by the authors in the introduction part, this work is suitable for publication in molecues, although it requires a thorough revision to correct typos and grammatical errors. Some of these errors are pointed out below for the authors’ consideration. Some sentences are also vague that need to be revised. A couple of examples and suggestions are also provided.
It would be more convenient if the chemical abbreviations such as TMS or TIPS in scheme 2 or TBAI in scheme 5, for example, could be defined in the text or in the captions. Please do this for all abbreviations.
- line 28 through 30 revise this sentence ”In this area, only a few of later transition metal catalysts, especially coin metals like gold, silver, platinum, and copper, could enable the activation of alkynes specially”
Suggestion: in this area, only a few metal complexes based on late transition metals such as platinum and coinage metals could enable the activation of alkynes.
- line 31: replace “which generated” with which are generated
- line 35: replace “industrial production” with industry
- line 39: replace “including” with include
- line 42: replace “The copper carbene species was first disclosed by Roy in 1906 [45], which was generated” with the copper carbene species, first identified by Roy et al. in 1906, were generated…
- line 43: delete “under”
- line 44: replace “is “ with was
- line 45-47: Revise this sentence:
“Until 2000, with abundant of copper salts were designed and ap-plied to the copper carbene chemistry, such as Cu(acac)2, Cu(hafaca)2, CuOTf, Cu(OTf)2, CuPF6, and others, a rapid growth has been enabled in copper carbene chemistry”
suggestion: “Until 2000, with abundant copper salts such as Cu(acac)2, Cu(hafaca)2, CuOTf, Cu(OTf)2, CuPF6, and others, suitable for the development of copper carbene chemistry, a rapid growth has been achieved in this area”
line 48: replace “the booming of ligand chemistry” with the advancement made in modern era ligand design
line 49: replace “especially” with specially
line 75: replace “moderated” with moderate
line 102: replace “products 14 were” with product 14 was.
Please note: Since 14 describes a general structure, therefore a verb describing a singular object need to be used even though it describes a collections of structures. This problem need to be fixed in other cases.
line 130: replace “products” with product
line 149: replace “which generated” with which was generated
line 170: replace “selectively” with selective
line 198: replace “groups” with group
line 198: add the letter “a” before compatible
line 210: replace “containing” with contained
This list goes on and on. So a careful revision is advised.
Author Response
For Reviewer 2:
The manuscript by Dong et al. describes two specific reactions of copper carbene species with alkynes, which includes their addition or cross couplings to carbenes. Since the topic presented in this review has not appeared in the literature before, as claimed by the authors in the introduction part, this work is suitable for publication in molecues, although it requires a thorough revision to correct typos and grammatical errors. Some of these errors are pointed out below for the authors’ consideration. Some sentences are also vague that need to be revised. A couple of examples and suggestions are also provided.
It would be more convenient if the chemical abbreviations such as TMS or TIPS in scheme 2 or TBAI in scheme 5, for example, could be defined in the text or in the captions. Please do this for all abbreviations.
Response: We thank the reviewer for pointing out. Additional table was added in the revised manuscript for the abbreviations
line 28 through 30 revise this sentence “In this area, only a few of later transition metal catalysts, especially coin metals like gold, silver, platinum, and copper, could enable the activation of alkynes specially”
Suggestion: in this area, only a few metal complexes based on late transition metals such as platinum and coinage metals could enable the activation of alkynes.
line 31: replace “which generated” with which are generated
line 35: replace “industrial production” with industry
line 39: replace “including” with include
line 42: replace “The copper carbene species was first disclosed by Roy in 1906 [45], which was generated” with the copper carbene species, first identified by Roy et al. in 1906, were generated…
line 43: delete “under”
line 44: replace “is“ with was
line 45-47: Revise this sentence:
“Until 2000, with abundant of copper salts were designed and ap-plied to the copper carbene chemistry, such as Cu(acac)2, Cu(hafaca)2, CuOTf, Cu(OTf)2, CuPF6, and others, a rapid growth has been enabled in copper carbene chemistry”
suggestion: “Until 2000, with abundant copper salts such as Cu(acac)2, Cu(hafaca)2, CuOTf, Cu(OTf)2, CuPF6, and others, suitable for the development of copper carbene chemistry, a rapid growth has been achieved in this area”
line 48: replace “the booming of ligand chemistry” with the advancement made in modern era ligand design
line 49: replace “especially” with specially
line 75: replace “moderated” with moderate
line 102: replace “products 14 were” with product 14 was.
Please note: Since 14 describes a general structure, therefore a verb describing a singular object need to be used even though it describes a collections of structures. This problem need to be fixed in other cases.
line 130: replace “products” with product
line 149: replace “which generated” with which was generated
line 170: replace “selectively” with selective
line 198: replace “groups” with group
line 198: add the letter “a” before compatible
line 210: replace “containing” with contained
This list goes on and on. So a careful revision is advised.
Response: We thank the reviewer for carefully pointing out these. Revisions have been made in the manuscript.
Reviewer 3 Report
Please see the attached document
-
Review on “Recent Advances in Catalytic Alkyne Transformation via Copper Carbene Intermediates” The paper by Dong et al. is written in improper English that unfortunately is not easy to correct because simply every single sentence is grammatically wrong. Moreover, English corrections are not sufficient because the chemistry meaning is not correctly expressed. I started with the first page and literally every single sentence is incorrect (please see below). I suggest to the authors to recruit an English-speaking chemist who can assist them with expressing correct the chemistry of the article. In the present form the article is unacceptable for publication.
Line 9 the word “which” refers to Earth the way the sentence is structured and the meaning is wrong then.
Line 10 Which advances?? The sentence before gives some facts about the element of Cu!
Line 11 “Copper carbenes”
Line 11 the word “which” again is place wrong!
Line 13 and 14 “a carbene intermediate”
Line 16 “placing emphasis”
Line 24 “alkynes have”
Line 24 sp should be a subscript
Line 26 “replace characters with characteristics
The sentence in line 26 is completely wrong, the characteristics have not led to properties but “render the properties etc.
Line 37 “much more appealing”
Line 38 “lower toxicity”, “easier accessibility”
Line 42 “a copper carbene”
I stop here corrections, because all the manuscript needs corrections line by line!
In addition: Figure 1 is not clear enough for publication.
The works discussed are selected in a partial way. I am not sure what the rational of the authors are in selecting specific representative works and not others. In particular, in some cases not the first advances are presented but random selections.
Author Response
Review on “Recent Advances in Catalytic Alkyne Transformation via Copper Carbene Intermediates” The paper by Dong et al. is written in improper English that unfortunately is not easy to correct because simply every single sentence is grammatically wrong. Moreover, English corrections are not sufficient because the chemistry meaning is not correctly expressed. I started with the first page and literally every single sentence is incorrect (please see below). I suggest to the authors to recruit an English-speaking chemist who can assist them with expressing correct the chemistry of the article. In the present form the article is unacceptable for publication.
Line 9 the word “which” refers to Earth the way the sentence is structured and the meaning is wrong then.
Line 10 Which advances?? The sentence before gives some facts about the element of Cu!
Line 11 “Copper carbenes”
Line 11 the word “which” again is place wrong!
Line 13 and 14 “a carbene intermediate”
Line 16 “placing emphasis”
Line 24 “alkynes have”
Line 24 sp should be a subscript
Line 26 “replace characters with characteristics
The sentence in line 26 is completely wrong, the characteristics have not led to properties but “render the properties etc.
Line 37 “much more appealing”
Line 38 “lower toxicity”, “easier accessibility”
Line 42 “a copper carbene”
I stop here corrections, because all the manuscript needs corrections line by line!
Response: We thank the reviewer for pointing out these out. Manuscript has been revised carefully according to these comments.
In addition: Figure 1 is not clear enough for publication.
Response: Thank you for pointing out, Figure 1 has been revised in manuscript.
The works discussed are selected in a partial way. I am not sure what the rational of the authors are in selecting specific representative works and not others. In particular, in some cases not the first advances are presented but random selections.
Response: We thank this reviewer for the comment. In this review, we mainly focus on the catalytic functionalization of alkynes involving copper carbene intermediates in the past decade. Two different reaction models have been summarized that take place between a copper carbene intermediate and alkyne species, including the cross-coupling reaction of copper carbene intermediate with terminal alkyne, and the addition of copper carbene intermediate onto the C-C triple bond. Other pioneering works or early examples related to copper carbene or alkyne activation has been briefly discussed only when closely related to above two reaction models. If there is any missing example, we would be more than pleasure to add to this review article.
Round 2
Reviewer 2 Report
none
Reviewer 3 Report
After the corrections, the article has improved.